# Could social prescribing contribute to type 2 diabetes prevention in people at high risk? Protocol for a realist, multilevel, mixed methods review and evaluation

Sara Calderón-Larrañaga ,[1,2] Megan Clinch,[1] Trisha Greenhalgh ,[3] Sarah Finer[1,4]

[1]Centre for Primary Care and Mental Health. Institute of Population Health Sciences, Barts and The London School of Medicine and Dentistry. Queen Mary University of London, London, UK
[2]Bromley By Bow Health Partnership, London, UK
[3]Nuffield Department of Primary Care Health Sciences, University of Oxford, Oxford, UK
[4]Barts Health NHS Trust, London, UK

**Correspondence to**
Dr Sara Calderón-Larrañaga;
s.calderon@qmul.ac.uk

## ABSTRACT

**Introduction** Social prescribing is an innovation being widely adopted within the UK National Health Service policy as a way of improving the management of people with long-term conditions, such as type 2 diabetes (T2D). It generally involves linking patients in primary care with non-medical community-based interventions. Despite widespread national support, evidence for the effectiveness of social prescribing is both insufficient and contested. In this study, we will investigate whether social prescribing can contribute to T2D prevention and, if so, when, how and in what circumstances it might best be introduced.

**Methods and analysis** We will draw on realist evaluation to investigate the complex interpersonal, organisational, social and policy contexts in which social prescribing relevant to T2D prevention is implemented. We will set up a stakeholder group to advise us throughout the study, which will be conducted over three interconnected stages. In stage 1, we will undertake a realist review to synthesise the current evidence base for social prescribing. In stage 2, we will investigate how social prescribing relevant to people at high risk of T2D 'works' in a multiethnic, socioeconomically diverse community and any interactions with existing T2D prevention services using qualitative, quantitative and realist methods. In stage 3 and building on previous stages, we will synthesise a 'transferable framework' that will guide implementation and evaluation of social prescribing relevant to T2D prevention at scale.

**Ethics and dissemination** National Health Service ethics approval has been granted (reference 20/LO/0713). This project will potentially inform the adaptation of social prescribing services to better meet the needs of people at high risk of T2D in socioeconomically deprived areas. Findings may also be transferable to other long-term conditions. Dissemination will be undertaken as a continuous process, supported by the stakeholder group. Tailored outputs will target the following audiences: (1) service providers and commissioners; (2) people at high risk of T2D and community stakeholders; and (3) policy and strategic decision makers.

**PROSPERO registration number** CRD42020196259.

### Strengths and limitations of this study

► Our study contributes to a much-needed evidence base for both social prescribing (SP) and its real-world implementation in specific patient groups, such as those at high risk of type 2 diabetes (T2D).
► The realist methodology will enable to evaluate existing SP programmes within the wider relational, organisational and policy contexts where they are delivered, including any potential interactions with existing National Health Service T2D prevention .
► We will seek participatory involvement from our stakeholders' advisory group throughout the study, which will strengthen the practical relevance and transferability of findings.
► The study does not intend to generate an effect size, but findings may guide the implementation and evaluation of SP initiatives relevant to T2D prevention at scale.

## INTRODUCTION

Social prescribing (SP) is an 'innovation' being widely adopted in the UK, including at policy level within the National Health Service (NHS) 'Long Term Plan'.[1 2] Although its definition varies, SP generally involves linking patients in primary care with community services offering employment, housing or financial advice, as well as a range of 'healthy lifestyle' activities, such as cooking classes, weight management or exercise programmes.[3] Health and lifestyle issues are supposed to be addressed alongside socioeconomic matters in an integrated way, usually with the help of a 'social prescriber' (also called 'link worker') who acts as link between health professionals and the wide range of community and voluntary services relevant to patients' situations.[4 5] Activities are typically non-medicalised, provided locally and

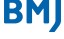

potentially more accessible and culturally appropriate to the local communities.[6]

Proponents of SP suggest it may improve health outcomes and redress health inequalities by addressing patients' social and behavioural determinants of health.[7] It is also argued that SP could improve the efficient use of NHS and social care resources by strengthening community networks and enhancing self-care.[8] However, current evidence base for SP is both scarce and contested. A recent systematic review published by Public Health England comprising eight UK studies (one cluster randomised controlled trial (RCT) and seven pre/post studies) found no clear evidence for the effectiveness of SP schemes.[9] In an earlier systematic review, Bickerdike and colleagues did not find enough evidence to prove either success or value for money across 15 studies.[10] Further systematic reviews on SP have failed to report consistent health, service utilisation or cost results.[11–15]

Authors suggest that the evidence gap on SP may be, in part, due to inadequate methodological approaches to gathering, assessing and/or synthesising data.[3] Because SP interventions have fuzzy boundaries, multiple and inter-related components and are extremely sensitive to the context, conventional research designs (namely RCTs and/or previous systematic reviews) may not be the most appropriate approach to evaluation.[16–18] Although RCTs would enable to compare the effectiveness of different variations of SP programmes, the methodological requirements to do so could impede the analysis of the interplay between programme implementation, the individuals involved, the programme's context and the wider social environment.[19] Novel methodological approaches are increasingly being used to address this 'complexity' and understand 'why', 'for whom' and 'in what circumstances' SP might (or might not) work.[5 20 21] However, no previous studies have evaluated SP in the context of interconnected interpersonal, organisational and policy relations, nor their influence on service delivery and implementation.

There is also a need to evaluate the role of SP in specific patient groups and populations.[22 23] This study will use type 2 diabetes (T2D) prevention as an exemplar and investigate the impact of SP in people at high risk of the condition based on the following considerations. First, T2D is a major public health concern, associated with reduced quality of life and life expectancy, and considerable socioeconomic consequences.[24] Although individual behavioural risk factors, such as physical inactivity and poor nutrition, are important determinants of T2D risk, these are heavily influenced and constrained by higher level social and demographic factors,[25 26] which appear to be major driving forces behind escalating T2D epidemics and health inequalities.[27 28] For T2D prevention strategies to be effective at population level and in the most high-risk groups, they most likely will need to address these social determinants in addition to focusing purely on individual-level behavioural interventions.[29] Second, current NHS delivered T2D prevention programmes,

such as 'Healthier You',[30] have not demonstrated high uptake.[31] The 'Healthier You' intervention consists of educational sessions over the course of nine months offered to people at high risk of T2D, but its effectiveness in preventing the condition remains uncertain.[32] Additionally, a social gradient exists in those engaging with, and gaining benefit from the 'Healthier You' programme.[33] There is significant concern that low uptake and high attrition may, at least in part, reflect the fact that individualised behavioural interventions do not adequately address critical social determinants of T2D.[34 35] Evaluating the impact of SP on a specific health area, such as T2D, will potentially enable to examine whether and how such interventions may be able to complement existing individualised behavioural approaches.

### Setting and context

This study will be based in Tower Hamlets, a multiethnic inner-city district in the East End of London, UK, where overall and T2D-related health outcomes have proved to be poorer compared with the national average[36] despite high-quality care.[37] Tower Hamlets is one of the most deprived boroughs in the UK and is characterised by its great ethnic diversity, with over half of all residents having a minority ethnic background.[38]

Local health intelligence data suggest that age-adjusted deaths directly attributable to diabetes are 8.93 per 100 000, compared with 5.06 across London and 5.06 across England.[37] Rates of T2D in Tower Hamlets are significantly higher among subpopulations of lowest socioeconomic status and South Asian population.[36 38 39] For instance, South Asians have a risk of developing T2D that is twofold greater than the local white population[40] and suffer from faster chronic kidney progression when they have diabetes.[41]

Tower Hamlets has become one of the 27 areas across the country chosen to be part of the first wave roll-out of the national NHS Diabetes Prevention Programme, 'Healthier You', and is also at the forefront of another innovative service model, SP, that is being rapidly adopted throughout the NHS. Tower Hamlets has a long-established local voluntary and community sector (VCS)[42] with a long history of delivering health promotion activities, including those relevant to T2D prevention. SP was recently rolled out borough-wide to strengthen and expand the partnership between the health and voluntary sectors to all local primary healthcare centres, building on the structure and experience developed in the pre-existing VCS and SP initiatives.[43] All patients registered with a Tower Hamlets general medical practice, aged over 18 and expressing a 'non-clinical' support need are eligible for the local SP programme. Each of the borough's 36 primary care settings is linked to at least one 'link worker', whose role ranges from signposting to more intensive approaches involving patients' needs assessments, ongoing support and recommendations of relevant VCS services.[43]

The proposed research will examine the existing evidence base for SP, and critically evaluate SP implementation and delivery in Tower Hamlets, using the following specific aims and questions.

### Aims

1. To investigate the possible impact of SP on people at high risk of T2D.
2. To inform the design, implementation and evaluation of SP initiatives relevant to people at high risk of T2D.

### Research questions

1. To what extent might SP meet the complex health and social needs of people at high risk of T2D in socioeconomically diverse populations?
2. What does 'good' practice in SP relevant to people at high risk of T2D look like and what are the main conditions ('active ingredients') for achieving this?
3. How do existing SP programmes address questions of implementation and evaluation, and what are the knowledge gaps critical to understanding their success or failure within a health system?

## METHODS

### Theoretical and conceptual framework

We will draw on realism, a theory-driven methodology that seeks to facilitate deep understanding of how complex interventions, such as SP, work and in what circumstances.[44 45] A central aspect of realism is its analysis of causation, which rejects the standard Humean model of regular succession of events. Realism takes the view that 'what causes something to happen has nothing to do with the number of times we observe it happening, instead identifying causal (also called 'generative') mechanisms of action and explaining how they work, whether they have been activated, and if so under what conditions'.[44] Our task as researchers will be to untangle the 'web of causation'[46] identified using this realist approach, and to critically illuminate the underlying structures and mechanisms that might shape and generate empirically observed outcomes.

In order to explore these underlying assumptions, realist methodology proposes the identification of context-mechanism-outcome configurations (CMOc).[47] A CMOc is a hypothesis that the programme works (or does not work) (O) because of the action of some underlying mechanisms (M), which only come into operation in particular contexts (C). 'If the right processes operate in the right conditions *then* the programme will prevail'.[48] These kinds of theoretical explanations, sufficiently abstract to draw transferable lessons but closed enough to the observable data to enable empirical testing and refinement, are referred to as 'programme theories'.[49]

CMOc may link to each other over time, with the outcome of one phase becoming the context of the next configuration in the chain of implementation steps (known as 'ripple effect').[50] Yet, CMOc may also organise concentrically.[48] In this study, we propose a multilevel

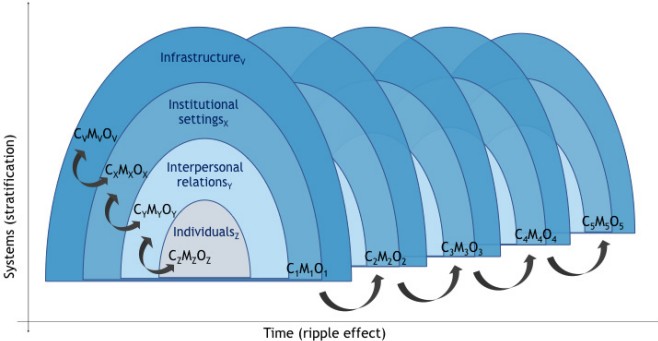

**Figure 1** Multilevel and dynamic realist framework. C, context; M, mechanism; O, outcome.

and dynamic realist framework (figure 1) to investigate the complex individual, interpersonal, organisational, social and policy contexts in which SP relevant to T2D is implemented, and illuminate key domains and tensions at these different and interconnected levels. A theory-testing strategy will be used to unpack these causal mechanisms and inter-related CMOc. Preliminary theoretical explanations will be made explicit and iteratively tested, confronted and refined using relevant and varied evidence from the primary literature (realist review[51]) and/or empirical data (realist evaluation[52]).

### Research plan

The study will be conducted in three interlinked stages over 3 years as follows:

- ► Stage 1—development and refinement of a realist programme theory for SP interventions relevant to people at high risk of T2D using a critical systematic literature review (realist synthesis).
- ► Stage 2—empirical testing and further refinement of a realist programme theory (guided by stage 1) in a socioeconomically deprived, ethnically diverse community at high risk of T2D using qualitative, quantitative and realist methods (realist evaluation).
- ► Stage 3—synthesis of secondary (stage 1) and primary (stage 2) evidence, involving the development of a transferable framework for guiding implementation and evaluation of SP relevant to T2D prevention at scale.

In parallel and as represented in figure 2, we will set up a stakeholder group comprising representatives of

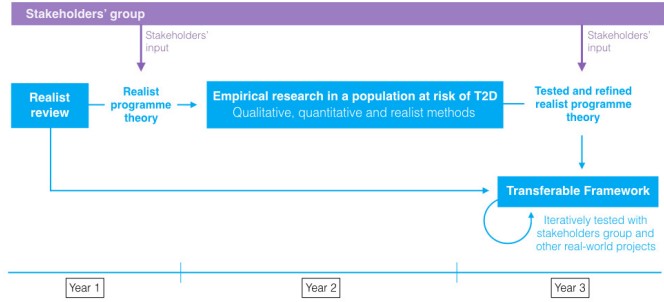

**Figure 2** Overview of the study design. T2D, type 2 diabetes.

the VCS, primary care leaders, members of Public Health England and relevant national charitable organisations (such as SP Network, Deep End Group and/or the National Academy for Social Prescribing). Via individual and group interviews, the study objectives and ongoing research outcomes will be iteratively shared, leading to discussions that will seek to explore and highlight stakeholders' opinions and concerns about SP, T2D prevention and the emerging programme theory.

### Stage 1. Development and refinement of a realist programme theory (realist synthesis)

Following the Realist And Meta-narrative Evidence Syntheses: Evolving Standards,[51 53] we will undertake a realist review in four iterative stages: (A) searching for evidence, (B) selecting and appraising literature, (C) extracting and organising data, and (D) synthesising the evidence and drawing conclusions.[54]

#### *Searching for evidence*

Searches will be organised and performed in two phases. The main search will be aimed at identifying the rationales for why and how SP might (or might not) play a role in T2D prevention and building preliminary programme theories that will be negotiated within the research and stakeholder groups. We will then undertake targeted searches aimed at refining these candidate programme theories.

We will use free text and indexing search terms employed in previous systematic reviews.[9 10 22] Terms for the second search will depend on the results from the first phase and will be discussed with a specialist librarian and the review team. We will search for qualitative, quantitative and mixed methods studies in relevant electronic databases and grey literature resources (the search strategy and databases for the main search are specified in online supplemental file 1). We will also undertake forward and backward citations chasing on included studies, hand searching of key journals and study author contact where appropriate.[55] Results will be exported to Rayyan QCRI for de-deduplication. The searching process will be reproduced by a second reviewer for consistency.

#### *Selecting and appraising articles*

Studies will be included if they consider adults (18+) in primary care settings and focus on interventions that link patients in primary care with community-based resources. Studies will be excluded if written in languages other than English, Spanish or French. Inclusion and exclusion criteria will continue to be refined throughout the review as theoretical understanding develops, and until theoretical saturation has been reached. Literature will be appraised based on their relevance at the point of inclusion ('does it contribute to theory building and testing?') and rigour ('are the methods used to generate the relevant data credible and trustworthy?').[53] A 10% random subsample of screening decisions will be reviewed by a second reviewer for consistency.

#### *Extracting and organising data*

The main reviewer will read all included articles, extract descriptive study characteristics and carry out manual coding. Full texts and the coding framework resulting from the manual analysis will then be uploaded into NVivo V.11 (qualitative data management software). Initial coding will comprise broad conceptual categories that will progressively be refined both inductively and deductively, and reframed applying a realist logic of analysis. This means that we will work iteratively to identify sections of the text related to contexts (C), mechanisms (M) and outcomes (O) and organise data in level-specific CMOc.[54] Again, a 10% random subsample of coded articles will be reviewed by a second reviewer for consistency, and disagreements will be solved by discussion.

#### *Synthesising the evidence and drawing conclusions*

We will use evidence synthesis processes proposed by realist methodology (eg, juxtaposition of data sources, reconciling 'contradictory' or disconfirming data, consolidation of sources of evidence)[48] to articulate clear and contextualised explanations on how and why certain patterns related to SP relevant to T2D prevention occur. These theory-informed, evidence-based, multilevel propositions will guide the empirical realist evaluation and the development of a transferable framework in stages 2 and 3, respectively.

### Stage 2. Empirical testing and further refinement of a realist programme theory (realist evaluation)
#### *Study design*

We will undertake a realist, multilevel and mixed methods evaluation of an existing SP programme based in Tower Hamlets, East London (the setting has been described above). The appropriateness and relevance of these approaches in the study and evaluation of complex, multidimensional, dynamic phenomena (such as SP) have been well established in the literature.[56–58]

At microlevel, we will characterise SP users and investigate their experience throughout the SP pathway (from initial referral to subsequent assessment by the link worker and unfolding community-based activities), paying special attention to the dynamic interaction between the different actors involved (patients, primary care clinicians, link workers, members of local community organisations, local authority/public health services). At mesolevel, we will explore how a primary care service and the VCS serving a community at high risk of T2D interact and operate to enable SP. We will also study the potential interaction of SP schemes with existing NHS T2D prevention programmes, such us 'Healthier You'. At macrolevel, we will investigate the barriers, facilitators and incentives to supporting SP relevant to T2D within the managerial and political sphere, and explore the interests and significances of SP for policy makers and other key stakeholders. Following a convergent mixed methods study design, qualitative and quantitative data will be collected and analysed 'in parallel' (during a similar time frame)

**Table 1** Overview of data source and collection in stage 2

| Data source | Data collection |
|---|---|
| SP service users | Interviews:<br>► Approximately 2 focus groups and 7 in-depth interviews.<br>► Purposive sampling, based on ethnicity, gender, age, diabetes risk, participation in SP.<br>Clinical data on all Tower Hamlets residents eligible for SP:<br>► Clinical measurements (HbA1c).<br>► Diagnosis codes (listed in QOF, including hypertension, diabetes, obesity).<br>► Disease risk (QRISK2, QDScore, pre-diabetes).<br>Sociodemographic data on all Tower Hamlets residents eligible for SP:<br>► Age and gender.<br>► Ethnicity.<br>► Ethnic density of local geography (postcode).<br>► Country of birth.<br>► First language spoken.<br>► Socioeconomic status (index of multiple deprivation, by lower layer super output area). |
| Primary care professionals, including link workers | Interviews:<br>► Approximately 2 focus groups and 7 in-depth interviews.<br>► Purposive sampling based on professional profile, type of contract, demographic characteristics (age, gender, ethnicity).<br>SP referrers' quantitative data:<br>► Demographic characteristics (age and gender).<br>► Primary care workers' professional profile and background.<br>► Type of contract. |
| VCS organisations | Interviews:<br>► Approximately 2 focus groups and 7 in-depth interviews.<br>► Purposive sampling based on type of activities, community embeddedness.<br>Observations of SP activities.<br>Key documentation: protocols, referral templates, advertising, web pages, and so on.<br>SP providers' quantitative data:<br>► Characteristics of the VCS organisations: available resources, areas of interest.<br>► Patients' attendance frequency. |
| SP service | Overview of the SP service:<br>► Existing referral criteria and reasons for referral.<br>► Number of referrals per network, surgery, health worker and profession.<br>Health economic data, including:<br>► Staff contact.<br>► Attendance rates.<br>► Facility use. |
| Policy makers and public health stakeholders | Interviews:<br>► Approximately 5 semistructured interviews.<br>► Snowball sampling.<br>Key documents and policy reports on SP. |

QOF, Quality and Outcomes Framework; SP, social prescribing; VCS, voluntary and community sector.

using an iterative approach, where data collection and analysis might drive changes in ongoing data collection procedures.[59 60]

### Data collection

Data will be generated through a number of qualitative and quantitative methods (table 1):

► Individual and group interviews. We will purposively select SP users living in an area at high risk of T2D (including those who did not turn up for appointments or declined to engage with the activities),

members of VCS organisations involved in SP, as well as local primary care clinicians, link workers and policy makers. Sampling will be guided by the strategies specified in table 1 and the findings of the realist review, as to the characteristics and circumstances that play a role in the success or failure of SP. Sample size will be determined by data saturation. An initial invitation letter and consent form will be provided and those wishing to participate will be contacted by a researcher. Interviews will be arranged at a time and

location convenient to the participants, and bilingual health advocates will be used where necessary.

► Observations of SP community-based activities. The main researcher will observe a sample of SP community-based activities, paying special attention to the interplay between the individual experience ('emic') and the sociocultural context ('etic') where these activities unfold.[61] Observational data on participants' behaviours, characteristics of activities, segments of dialogue between actors and contextual features will be kept in the form of field notes for ulterior analysis.

► Documents. Relevant local SP documentation and policy reports will be accessed for further analysis. We will ask each interviewed policy maker to reference key documents that might be guiding policy in this area.

► Quantitative individual-level data. We will characterise the clinical and sociodemographic features of SP users and the background population eligible for SP (including users of the local 'Healthier You' programme), using anonymised data from high-quality primary care electronic health records (available from the Clinical Effectiveness Unit, Queen Mary University of London (QMUL)) and local authority registers.

► Quantitative service-level data. We will gather SP referrers' data by accessing primary care case management systems (Egton Medical Information Systems, EMIS) and/or referral forms. Quantifiable data of local public, private and non-profit institutions and citizens' associations will also be accessed using link worker's referral forms and the records of the organisations under study. We will collect SP costs data to test the feasibility of a future scaled-up evaluation of SP and T2D prevention frameworks and their potential impact on health systems.

### Data analysis

Qualitative data will be analysed thematically to identify the patterns and theoretical explanations derived from its concepts, categories and inter-relations.[62] Data will be analysed according to particular features, but without losing sight of the 'big picture' and the contexts in which they arise.[63] In combination with thematic analysis, we will apply a realist logic of analysis to the data to complement, confirm, confront and/or refine the programme theory deriving from the review. Data analysis and interpretation will be reflexively monitored.[64] We will seek negative cases and triangulate within the research group to incorporate and maintain validity, and hence ensure rigour in the research process.[65 66]

Quantitative data will be used to characterise the potential impact of SP on people at high risk of T2D, and in the context of other existing T2D prevention strategies. These data will not provide a definitive evaluation of SP, but will provide important context for the qualitative research findings and will indicate areas of focus and choice of

endpoints (and their feasibility) for future at-scale evaluation. Data will be analysed using descriptive statistics to characterise the clinical and sociodemographic features of patients accessing SP. Comparative statistical tests will be used to look for differences in the SP recipient population compared with the total background population aged 18+ (the eligibility criteria for SP are age 18+ and a subjective assessment of need), and to individuals participating in the NHS Type 2 Diabetes Prevention Programme, 'Healthier You'. Multivariate analysis will enable the identification of potential factors that might be associated with the likelihood of being referred into SP and characterise the uptake and equity of access to the programme across the borough. Considering that 2% of the population receives SP and 1 in 11 people in Tower Hamlets are at high risk of developing T2D (QDScore >20), we anticipate having sufficient power (0.8) to detect a 0.7% absolute difference (35% relative difference) in the likelihood of receiving SP among diabetes risk groups, using a type 1 error rate of 0.05.

### Synthesis and integration of data

Qualitative findings, framed within CMOCs, will be synthesised according to microlevel, mesolevel and macrolevel (and interconnections between levels) that most accurately represent how and why SP programmes relevant to T2D prevention might (or might not) 'work'. Quantitative outcomes will be used to provide deeper context to the multilevel programme theory. We will undertake a narrative integration of findings[60] and use visual means[67] where appropriate.

### Stage 3. Development of a transferable framework to guide implementation and evaluation of SP relevant to T2D

Building on the findings of the critical systematic literature review (stage 1) and the empirical research (stage 2), we will synthesise a transferable framework for understanding and evaluating SP programmes relevant to people at high risk of T2D. In order to enhance generalisability, a near final version of the framework will be shared with stakeholders involved in SP programmes relevant to T2D nationwide and further refined based on their inputs.

Similar approaches, such as the NASSS framework (for theorising and evaluating non-adoption, abandonment, scale-up, spread and sustainability of health and care technologies),[68] have previously been successfully developed and used in the field of health services research and complex interventions, and have the potential to translate research findings to clinical care, reducing the 'research-implementation divide' that research on complex interventions frequently encounters.

### Ethics and dissemination

Health Research Authority and Health and Care Research Wales ethics approval has been granted (reference 20/LO/0713). Standard rules apply for data security, confidentiality and information governance. We will seek

informed consent for observations during community-based interventions and interviews. We will protect the confidentiality and privacy of participants in focus groups by requesting acceptance of a code of conduct to ensure that personal and private information is not shared outside the group.

Dissemination of study outputs will be undertaken as a continuous process and be facilitated by the stakeholder group. We anticipate that the transferable framework developed in stage 3 will constitute a key output. We will develop workshops, presentations and summary documents to make it accessible to potential users. Senior policy contacts, who are represented on our stakeholder group, will be ideally placed to facilitate its wide-reaching dissemination and implementation. We plan outputs for four main audiences.

► For service providers (including primary care clinicians, link workers and the VCS), we will provide guidance on how to evaluate and improve local SP schemes.

► For people living in areas at high risk of T2D and community stakeholders, we will develop lay summaries and user-friendly versions of all our findings.

► For policy makers and strategic decision makers, we will produce succinct and accessible briefing papers of key findings aimed at informing prevailing policy and commissioning decisions.

► For the academic community, we will produce research publications in peer-reviewed, open-access journals and conference presentations. The whole proposed research will constitute the core contents of a PhD thesis, whose submission and defence will be accomplished from the QMUL.

## Patient and public involvement

Patient and public involvement is central to the development of this project and for the plans on how it will be executed. To date, this involvement has come from several groups, including local East London community members (through pilot work to identify the research questions, scope and acceptability), the QMUL-Barts Health Diabetes Lay Research Panel, as well as using Diabetes UK resources including the recent James Lind Alliance T2D priority setting partnership.[69] As previously explained, our 'stakeholder advisory group' will also help to enhance the practical relevance and impact of our research.

## DISCUSSION

There is an increasing interest in SP as a means of addressing the wider social determinants behind escalating burden of long-term conditions, such as T2D, and health inequalities. As part of the NHS Long Term Plan,[1] NHS England plans to recruit 1000 SP link workers so that the service is available in every general medical practice by 2024.[70]

Despite widespread policy support and proliferation, evidence for the effectiveness of SP is currently sparse, its implementation is heterogeneous and complex, and it incorporates little robust evaluation across health domains.[71] Some of the recent studies on SP have used novel methodological approaches to better understand 'how', 'why' and 'in what circumstances' it might or (might not) work.[4 20 21] Qualitative research studies have also provided valuable insight into the processes of programme implementation and delivery.[72–75] However, no previous studies have addressed and evaluated SP in the context of dynamic, interconnected, multilevel systems. Nor have they 'critically' analysed this context to purposively unfold the tensions, contradictions and competing interests that can potentially shape service delivery.

This study brings a novel methodological approach to investigate whether SP may contribute to comprehensive, community-embedded T2D prevention strategies in primary care and its potential impact on patients at high risk. Our multilevel analytical approach will potentially illuminate the complexity of SP and the system in which it operates (including interpersonal, organisational, policy contexts). It is envisaged that project outcomes will inform the design, implementation and evaluation of SP initiatives relevant to T2D by understanding when, how and in what circumstances this service model might best be introduced.

**Correction notice** The corresponding author name has been corrected to Sara Calderón-Larrañaga.

**Contributors** SCL and SF conceptualised the study with input from MC and TG. SCL wrote the first draft of this manuscript with significant input from SF, MC and TG. All authors have read and approved the final manuscript.

**Funding** This work was supported by the Economic Social Research Council, London Interdisciplinary Social Sciences Doctoral Training Partnership (grant number ES/P000703/1) and Currier's Millennium Healthcare Bursary.

**Disclaimer** The views and opinions expressed therein are those of the authors and do not necessarily reflect those of the London Interdisciplinary Social Sciences Doctoral Training Partnership, Currier's Millennium Healthcare Bursary, National Health Service or the Department of Health.

**Competing interests** None declared.

**Patient consent for publication** Not required.

**Provenance and peer review** Not commissioned; externally peer reviewed.

**ORCID iDs**
Sara Calderón-Larrañaga http://orcid.org/0000-0003-1369-0250
Trisha Greenhalgh http://orcid.org/0000-0003-2369-8088

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
