## [Reviewer comments · BMJ Open]

ARTICLE DETAILS

TITLE (PROVISIONAL)	Could social prescribing contribute to type 2 diabetes prevention in people at high risk? Protocol for a realist multi-level, mixed-methods review and evaluation.
AUTHORS	Calderón Larrañaga, Sara; Clinch, Megan; Greenhalgh, Trisha; Finer, Sarah

VERSION 1 – REVIEW

REVIEWER	Yannis Pappas Institute for Health Research, University of Bedfordshire, UK
REVIEW RETURNED	28-Jul-2020

GENERAL COMMENTS	Thank you for sharing this study protocol with me. It is true that social prescription interventions, especially because of their wide use, need to be based on evidence. It is therefore important that small and large studies contribute to that end. The protocol outlines an ambitious study which aims to make a theoretical as well as a practical contribution to the field. Please consider the following comments: 1. In what sense is social prescription an innovation? According to which definition is considered one? At which point the convergence of existing interventions become an innovation? Even if it was once an innovation, after several years in operations, multiple iterations and delivery methodologies can still be considered an innovation? I mention this because whether SP is defined as innovation or an intervention, might have consequences on the theoretical work that is being proposed.2. How is the suggested review of the literature different to current ones and what it is contributing?3. Will this be a systematic review? If yes, please be clear every time you refer to this. It appears that the protocol is shying away at the moment.4. I suggest that a search strategy for the systematic review is provided in this protocol.5. A justification of why only studies written in English, Spanish and French are included in the review.6. My most substantial concern is how will aim 2: 'To inform the design, implementation and evaluation of SP initiatives relevant to people
---

	at high risk of T2D' be met? Will the data from Tower and Hamlets, a diverse but deprived Borough, be generalisable enough to meet the most ambitious aim of the study? 7. The previous point brings me to the limitations of the study, which need to be discussed liberally, not least in relation to the capacity of the study design and sample to make a To inform the design, implementation and evaluation of SP initiatives relevant to people at high risk of T2D
--	--

REVIEWER	Josephine Wildman Newcastle University, Population Health Sciences Institute, UK
REVIEW RETURNED	10-Aug-2020

GENERAL COMMENTS	Thank you for the opportunity to review this protocol for a realist evaluation of social prescribing in Tower Hamlets. The study seeks to answer some interesting and important questions. In light of this, the protocol needs to be clearer about what the study seeks to do. For example, in the abstract (and other places in the text), the authors' state that the study will investigate whether social prescribing can contribute to T2D prevention; however, in other places the study stresses that it will not look at effectiveness in terms of outcomes but rather will characterise the intervention, its components, the interplay between them and the mechanisms by which it operates - with a view to evaluating future effectiveness. All these are important questions. The authors are probably familiar with the MRC's guidance on evaluating complex interventions (https://mrc.ukri.org/documents/pdf/complex-interventions-guidance/). What this protocol proposes will address some important aspects of this process and would be useful to reference (apologies if this has been done, I couldn't see it in the ref list). Particularly important questions that the proposed study will address are who is referred and why do they engage (or not)? The authors should highlight this contribution more clearly, not least because of the risk in all interventions of widening health inequalities if people most at need are not reached. Also of importance is the study's aim of exploring the interactions between primary care, the VCSE and commissioners. Again, this contribution should be highlighted. The exploration of the interaction between SP and the Healthier You programme is an important aspect of this study that should be highlighted. Questions of duplication of effort and efficient use of resources in the implementation of the SP roll-out need addressing. The intervention: Much more detail about the SP intervention is needed. Including (but not limited to): Who is recruited (eligibility criteria) and how? What ingredients make up the intervention? How long does it last? Does it target people at risk of T2D? Does the intervention seek to prevent T2D or aim to help people who already have T2D? Although this distinction is made in the paper, the outcomes include HbA1c, which (I think) would only be measured in patients who have T2D. I was left with a number of questions about the proposed methods: SP service users: More detail is needed on a range of points, including: what is the rationale for the number of interviews/focus groups? The protocol just says "sufficient sample", but how will this be judged? How will potential participants be identified? Are there gatekeepers? How does the study propose to engage those who
---

	have been reluctant to take part in SP? How will the study recruit those beyond the 'usual suspects' who take part in research? Why focus groups? Given that this is a multi-ethnic community, more detail is needed on how the study proposes engaging with people whose first language is not English, who may have literacy problems etc? Where will the interviews take place? Make it clearer that sampling will be guided in part by the evidence review. More detail is needed on the quantitative data. What is the Clinical Effectiveness Unit, QMUL? For what purpose does it collect data? What data does it collect? How does it relate to QOF data? Who can access the data? How are users of Healthier You to be identified? How and where is participation in SP recorded? What is the quality of this data? Given the use of identifiable patient-level data proposed in this protocol, I would like an explicit statement that permission has been granted and processes are in place to allow the study team to access patient data. Also, are the clinical measurements confined to BMI and HbA1c? Which QOF diagnosis codes are relevant? Have data minimisation requirements been addressed? Where is the socio-demographic data on all Tower Hamlets residents eligible for SP coming from? Primary care and link workers: Which primary care worker roles are to be included? Why is type of contract interesting? VCS organisations: From what will these be sampled? E.g. is there a database? How will community embeddedness be judged? Caution is required in the use of the term 'ethnographic'. If the research method is truly ethnographic rather than observation, more detail is needed about what this will entail. SP service: Will health economic data include costs? Finally, the table could be better employed; e.g. perhaps organised by data subject, data source (including details of what it is, how accessed etc.), variables (list of all to be used)/sampling criteria. Minor points: Is 'social prescriber' a well-used synonym for a facilitator? Link worker or community health worker are perhaps more familiar terms? Will the realist literature review be confined to high-income countries? There are a few typos that need correcting.
--	---

VERSION 1 – AUTHOR RESPONSE

Reviewer 1

1. In what sense is social prescription an innovation? According to which definition is considered one? At which point the convergence of existing interventions become an innovation? Even if it was once an innovation, after several years in operations, multiple iterations and delivery methodologies can still be considered an innovation? I mention this because whether SP is defined as innovation or an intervention, might have consequences on the theoretical work that is being proposed.

Thank you very much for raising this relevant point. For the purposes of this study, we draw on the definition proposed by Greenhalgh et al. in their seminal study 'Diffusion of innovations in service organisations: systematic review and recommendations'(1). The full, detailed

review also came out as a book in 2005(2). We believe that Social Prescribing is in keeping with the definition proposed by the authors, based on the following considerations:

- The application of SP to the context of type 2 diabetes prevention constitutes a set of routines and ways of working that are perceived as new by stakeholders
- It is linked to the provision or support of healthcare
- It is directed at improving health outcomes, administrative efficiency, cost-effectiveness, or the user experience
- It is implemented by means of planned and co-ordinated action by individuals' teams or organisations

This definition embraces (but is not limited to) educational, lifestyle or other complex interventions, including the introduction of new staff roles (e.g., link workers or social prescribers).

2. How is the suggested review of the literature different to current ones and what it is contributing?

Although some recent studies on SP have used realist methods to better understand 'how', 'why' and 'in what circumstances' it might or (might not) work(3,4), no previous reviews have evaluated SP in the context of dynamic, interconnected, multilevel systems. Nor have they analysed its potential impact on areas of specific health need, such as type 2 diabetes prevention.

As highlighted in the 'discussion' section of our protocol paper: 'this study brings a novel methodological approach to investigate whether SP may contribute to comprehensive, community embedded T2D prevention strategies in primary care and its potential impact on populations at high risk. Our multilevel analytic approach will potentially illuminate the complexity of SP and the system in which it operates (including interpersonal, organisational, policy contexts'.

3. Will this be a systematic review? If yes, please be clear every time you refer to this. It appears that the protocol is shying away at the moment.

We would like to thank the reviewer for this valuable observation. Our review will be developed systematically, following rigorous methodological guidance for realist reviews as described in the RAMESES quality standards(5) and has been registered with PROSPERO (CRD42020196259). We have introduced the following modifications to the main text in order to make this point clearer:

- In the 'methods' section, 'research plan' subsection: Stage 1 – Development and refinement of a realist programme theory for SP interventions relevant to populations at high risk of T2D using a critical systematic literature review (realist synthesis).
- In the 'methods' section, 'Stage 3' subsection: Building on the findings of the critical systematic literature review (Stage 1) and the empirical research (Stage 2), we will synthesise a transferable framework for thinking, understanding and evaluating SP programs relevant to populations at high risk of T2D.

4. I suggest that a search strategy for the systematic review is provided in this protocol.

Following your comment, we have included the precise search strategy for each database in Appendix 1 and modified the main text as follows:

In 'methods' section, 'Stage 1' subsection: We will search for qualitative, quantitative and mixed-methods studies in relevant electronic databases and grey literature resources (the search strategy and databases for the main search are specified in Appendix 1).

5. A justification of why only studies written in English, Spanish and French are included in the review.

The Cochrane Handbook acknowledges the risk of bias in reviews containing exclusively English language studies(6). Similarly, the methodological guidelines for Campbell Collaboration reviews warn against the risk of language bias and encourages authors not to restrict by language(7). The fact that the main researcher in our review team is a fluent French and Spanish speaker allowed us to broaden the scope of our searches and also consider articles written in French and Spanish.

It is, however, worth noting that only two of the 1776 records identified through database searching had been written in languages other than English. It is, therefore, unlikely that language-based inclusion criteria have had any impact on the findings of this systematic review.

6. My most substantial concern is how will aim 2: 'To inform the design, implementation and evaluation of SP initiatives relevant to people at high risk of T2D' be met? Will the data from Tower and Hamlets, a diverse but deprived Borough, be generalisable enough to meet the most ambitious aim of the study? The previous point brings me to the limitations of the study, which need to be discussed liberally, not least in relation to the capacity of the study design and sample to 'inform the design, implementation and evaluation of SP initiatives relevant to people at high risk of T2D'

We appreciate the reviewer's relevant concerns regarding study generalisability. Our study has been carefully designed in order to overcome such limitations and inform SP best practice across NHS primary care in the UK, using the following approaches:

1. We are undertaking a broad systematic realist review comprising approximately 140 studies from diverse settings and contexts.
2. The realist methodology will enable to understand 'why', 'how' and 'in what circumstances' these interventions may (or may not) work, by elucidating mechanisms that can be applied and be relevant to different contexts.
3. Our empirical research based in Tower Hamlets will be supported by continuous involvement of a stakeholder advisory group that comprises both local and national representatives from provider and policy-based organisations, e.g. Public Health England and relevant national charitable organisations (such as SP Network, Deep End Group, and/or the National Academy for Social Prescribing). The expertise from these nationally-representative stakeholders will support the testing and refinement of our Transferable Framework, and make findings generalisable and relevant to end user.

Reviewer 2

1. In the abstract (and other places in the text), the authors' state that the study will investigate whether social prescribing can contribute to T2D prevention; however, in other places the study stresses that it will not look at effectiveness in terms of outcomes but rather will characterise the intervention, its components, the interplay between them and the mechanisms by which it operates - with a view to evaluating future effectiveness.

We would like to thank the reviewer for this observation. Following MRC guidance on the evaluation of complex interventions(8), we will undertake a 'Process Evaluation' (as opposed to simple outcome evaluations in isolation) in order to gather meaningful and contextually rich evidence of real-world

SP schemes. Our aim, therefore, is not to evaluate programme effectiveness. This process evaluation will be underpinned by a realist methodology to help understand and identify the specific conditions and mechanisms (the “how, when and in what circumstances”) that may shape service delivery.

2. The authors are probably familiar with the MRC's guidance on evaluating complex interventions (<https://mrc.ukri.org/documents/pdf/complex-interventions-guidance/>). What this protocol proposes will address some important aspects of this process and would be useful to reference (apologies if this has been done, I couldn't see it in the ref list).

Thank you very much for noting our omission. We have now included the reference.

3. Particularly important questions that the proposed study will address are who is referred and why do they engage (or not)? The authors should highlight this contribution more clearly, not least because of the risk in all interventions of widening health inequalities if people most at need are not reached.

We strongly agree with the reviewer on the importance of studying (non)attendance and/or (non)engagement, in order to design interventions accessible to those in greatest need. We have modified the section ‘Empirical testing and further refinement of a realist programme theory (Realist Evaluation)’ as follows in order to highlight that this will be included in our analyses:

‘Multi-variate analysis will enable to identify potential factors that might be associated with the likelihood of being referred into SP and characterise the uptake and equity of access to the programme across the borough.’

4. The exploration of the interaction between SP and the Healthier You programme is an important aspect of this study that should be highlighted.

We are grateful for the reviewer's observation and have included the following sentence in the section ‘Strengths and Limitations of this study’ in order to highlight this important aspect:

‘The realist methodology will enable to evaluate existing SP programmes within the wider relational, organisational and policy context where they are delivered, including any potential interactions with existing NHS T2D prevention services’

5. The intervention: Much more detail about the SP intervention is needed. Including (but not limited to): Who is recruited (eligibility criteria) and how? What ingredients make up the intervention? How long does it last?

All patients registered with a Tower Hamlets GP, aged over 18 and expressing a ‘non-clinical’ support need are eligible for the local SP programme. The link between the health and the third sectors is provided by a ‘social prescriber’ (also called ‘link worker’), whose role ranges from signposting to more intensive approaches involving patients’ needs assessments, ongoing support and recommendations of relevant VCS services. The duration of the service is variable depending on patients’ needs and circumstances.

We have included the following sentence in the ‘Setting and context’ section in light of your comment:

‘SP is being rolled out borough-wide by the Clinical Commissioning Group, aimed at strengthening and expanding the partnership between the health and voluntary sectors to all local Primary Health Care centres, building on the structure and experience developed in the pre-existing VCS and SP schemes(40). All patients registered with a Tower Hamlets GP, aged over 18 and expressing a ‘non-clinical’ support need are eligible for the local SP programme. Each of the borough’s 36 primary

care settings is linked to at list one 'link worker', whose role ranges from signposting to more intensive approaches involving patients' needs assessments, ongoing support and recommendations of relevant VCS services. (40).'

6. Does it target people at risk of T2D? Does the intervention seek to prevent T2D or aim to help people who already have T2D? Although this distinction is made in the paper, the outcomes include HbA1c, which (I think) would only be measured in patients who have T2D.

The SP service is not specifically designed for patients with or at high risk of T2D. As specified above, eligibility is contingent upon being registered with a Tower Hamlets GP, aged over 18 and expressing a 'non-clinical' support need.

However, T2D risk is heavily influenced and constrained by higher-level socio-economic factors (unemployment, food insecurity, etc.) that could potentially be tackled with SP. In addition, some of the community-based activities accessed via SP are lifestyle-related (such as, exercise, healthy eating and weight management) and, hence, relevant to T2D prevention.

Measurement of Hb1Ac is used for the identification of individuals at risk of type 2 diabetes (e.g. HbA1c between 42 to 47 mmol/mol denotes 'prediabetes'). 'Prediabetes' is a critical stage in the development of T2D and is one of the referral criteria for the NHS 'Healthier You' diabetes prevention programme.

7. More detail is needed on a range of points, including: what is the rationale for the number of interviews/focus groups? The protocol just says "sufficient sample", but how will this be judged? Make it clearer that sampling will be guided in part by the evidence review.

We have provided an approximate number of interviews and focus groups in table 1. However, the number of contacts and interviews will be guided iteratively during the research, and until data saturation has been reached(9).

In light of the reviewer's observation and comment, we have modified the text as shown below:

'We will purposively select SP users living in an area at high risk of T2D (including those who did not turn up for appointments or declined to engage with the activities), members of VCS organisations involved in SP, as well as local primary care clinicians, social prescribers and policy makers. Sampling will be guided by the strategies specified in table 1 and the findings of the realist review, as to the characteristics and circumstances that play a role in the success or failure of SP. Sample size will be determined by data saturation. An initial invitation letter and consent form will be provided and those wishing to participate will be contacted by a researcher. Interviews will be arranged at a time and location convenient to the participants, and bilingual health advocates will be used where necessary.'

8. How will potential participants be identified? Are there gatekeepers? How does the study propose to engage those who have been reluctant to take part in SP? How will the study recruit those beyond the 'usual suspects' who take part in research?

Thank you for raising these relevant concerns.

Primary care workers (including link workers), during routine consultations, will ask eligible patients if they want to take part. They will provide an initial invitation letter and Participant Information Sheet. Those wishing to participate will be contacted by the researcher (by post, phone or email), who will

provide further research details if required. As for the referring primary care workers, potential interviewees will be directly contacted by the researcher (by post, phone or email).

Recruitment will be facilitated by our stakeholders advisory group, who have great knowledge of the local community, primary care networks and SP scheme. Link workers will also be key for the effective recruitment of potential participants and will be able to provide relevant information about SP users and community-based service providers. We will work closely with them to identify and recruit a representative sample of local VCS organisation and SP users, including those who have been referred into SP but have failed to engage in the prescribed activities. Lastly, recruitment will also be facilitated by the fact that the primary investigator is a GP working in the area under study, with relevant contacts with other primary care workers and community organisations.

9. Given that this is a multi-ethnic community, more detail is needed on how the study proposes engaging with people whose first language is not English, who may have literacy problems etc?

I would like to thank the reviewer for raising this important point. I strongly agree that literacy and digital exclusion issues need to be considered in order to effectively engage with non-English speakers. As specified in the 'methods/data collection' section, bilingual health advocates will be used where necessary in order to ensure appropriate recruitment and communication with participants from minority ethnic groups. It may also be worth highlighting that there is a long history within this health system of working effectively across cultural and language barriers and using health advocates for both research and clinical practice. In addition, some of the link workers are from relevant ethnic backgrounds in the local community, which will support crossing cultural and language barriers effectively.

10. Primary care and link workers: Which primary care worker roles are to be included? Why is type of contract interesting?

Referrals to the local SP scheme are encouraged from all professionals within primary care, which may include general practitioners, practice nurses, pharmacists, health care assistants, practice managers or members of the reception team. The characterisation of the referring professionals, including type of contract (partner, salaried, locum), will enable to identify factors associated with referral rates.

11. Where will the interviews take place?

The interviews will be arranged by study researchers at a time and location convenient to the participants. They will be conducted in person, by phone or online depending on participants' preference, and in compliance with Covid19-related safety recommendations.

Following your comment, we have extended the 'methods/data collection' section further as follows:

'We will purposively select SP users living in an area at high risk of T2D (including those who did not turn up for appointments or declined to engage with the activities), members of VCS organisations involved in SP, as well as local primary care clinicians, link workers and policy makers. Sampling will be guided by the strategies specified in table 1 and the findings of the realist review, as to the characteristics and circumstances that play a role in the success or failure of SP. Sample size will be determined by data saturation. An initial invitation letter and consent form will be provided and those wishing to participate will be contacted by a researcher. Interviews will be arranged at a time and location convenient to the participants, and bilingual health advocates will be used where necessary.'

12. VCS organisations: From what will these be sampled? E.g. is there a database? How will community embeddedness be judged?

We will choose a sample of VCS organisation based on the type of activities they offer and their degree of community embeddedness, which refers to the extent to which organisations are ingrained or fixed in the local community. Identification and recruitment will be facilitated by the stakeholders advisory and group and local link workers, who have great knowledge of the community and existing third sector organisations. Link workers have developed (and will share with us) an updated inventory/database of local community resources where patients can be referred into. We will work closely with them to characterise local VCS organisations and recruit a relevant and representative sample for our study.

13. More detail is needed on the quantitative data. What is the Clinical Effectiveness Unit, QMUL? For what purpose does it collect data? What data does it collect? How does it relate to QOF data? Who can access the data?

The Clinical Effectiveness Group (CEG) is a unit based at Queen Mary University of London (QMUL), who provide access to high quality, anonymised data from primary care electronic health records (including QOF data).

CEG provide locally relevant guidelines, in-practice facilitation, clinical tools, clinical templates and health informatics for commissioning and clinical management. They also lead research in the effective delivery of primary care and are a partner in the Discovery programme which integrates hospital, GP, and local authority records, and in both the pan-London MRC Health Data Science UK and the pan-London Local Integrated Health Record Exemplar (for more information, please visit <https://www.qmul.ac.uk/blizard/ceg/>).

For the purpose of this study, we have already liaised with the CEG and secured their support and supervision to access and analyse relevant clinical and sociodemographic data of SP users (as specified in table 1).

14. How are users of Healthier You to be identified? How and where is participation in SP recorded? Given the use of identifiable patient-level data proposed in this protocol, I would like an explicit statement that permission has been granted and processes are in place to allow the study team to access patient data.

Referrals to both 'Healthier You' and SP are recorded in the primary care case management systems (EMIS). This data is anonymised at source and NOT linked with individual participants' data from the qualitative studies. It may also be worth highlighting that our study has secured HRA and Health and Care Research Wales (HCRW) Approval, on the basis described in this protocol.

In light of your concerns and comment, we have included the following clarification in the 'data collection' section:

'Quantitative individual-level data. We will characterise the clinical and socio-demographic features of SP users and the background population eligible for SP (including users of the local 'Healthier You' programme), using anonymised data from high quality primary care electronic health records (available from the Clinical Effectiveness Unit, Queen Mary University of London) and local authority registers.'

15. Also, are the clinical measurements confined to BMI and HbA1c? Which QOF diagnosis codes are relevant? Have data minimisation requirements been addressed?

Thank you for raising this question which on reflection, we had left unclear in our table. Clinical measurements will be confined to HbA1c, which will enable to identify patients with prediabetes. Other clinical measurements associated with risk of future T2D (including BMI, blood pressure, smoking status, and so on) are already comprised within the validated T2D risk algorithm, QDScore. As for QOF diagnostic codes, we will consider, 'Hypertension', 'Diabetes' and 'Obesity'. Following the data minimisation principle, we will only collect and process data that is adequate, relevant and limited to what is necessary for the purpose of this research. We have updated table 1 in accordance with these considerations.

16. Where is the socio-demographic data on all Tower Hamlets residents eligible for SP coming from?

Socio-demographic data set will be drawn from electronic general practice records and local authority registers. These will include different social categories and axes of difference or disadvantage, such as age and gender, ethnicity, ethnic density of local geography (postcode), country of birth, first language spoken, socio-economic status (estimated using Index of Multiple Deprivation, by Lower Super Output Area).

17. Caution is required in the use of the term 'ethnographic'. If the research method is truly ethnographic rather than observation, more detail is needed about what this will entail.

Following your relevant observation, we have removed the term 'ethnographic' from the manuscript to avoid any misunderstanding.

18. SP service: Will health economic data include costs?

This study is not designed or powered to undertake a detailed health economic analysis, but will be used to test feasibility of evaluating a future scaled up evaluation of SP and diabetes prevention frameworks and their potential impact on health systems. As specified in the research proposal, we will collate data on resource use (e.g. staff contact, attendance rates, use of facilities) in order to characterise the current costs of delivering the scheme to the NHS, as well as implications with regards uptake, persistence with and use of other care.

19. Finally, the table could be better employed; e.g. perhaps organised by data subject, data source (including details of what it is, how accessed etc.), variables (list of all to be used)/sampling criteria.

We thank the reviewer for the suggestion. We have modified the table in accordance with your previous valuable comments and observations, which we believe have substantially enhanced its accuracy and clarity.

20. Is 'social prescriber' a well-used synonym for a facilitator? Link worker or community health worker are perhaps more familiar terms?

Following your observation, we have replaced the term 'social prescriber' with 'link worker' throughout the manuscript.

21. Will the realist literature review be confined to high-income countries?

The diversity of study settings and contexts could potentially enhance the generalisability of our realist review findings. We have not therefore preestablished any restrictions based on countries' income and included all studies on interventions linking adults (>18) in primary care with VCS organisations, regardless of study design (quantitative, qualitative and mixed methods) or country of origin.

REFERENCES:

1. Greenhalgh T, Robert G, Macfarlane F, Bate P, Kyriakidou O. Diffusion of Innovations in Service Organizations: Systematic Review and Recommendations. *Milbank Q* [Internet]. 2004 Dec;82(4):581–629.
2. Greenhalgh T, Robert G, Bate P, Macfarlane F, Kyriakidou O. Diffusion of Innovations in Health Service Organisations: a systematic literature review. Oxford: Blackwell Publishing Ltd; 2005.
3. Tierney S, Wong G, Roberts N, Boylan A-M, Park S, Abrams R, et al. Supporting social prescribing in primary care by linking people to local assets: a realist review. *BMC Med* [Internet]. 2020 Dec 13;18(1):49.
4. Husk K, Blockley K, Lovell R, Bethel A, Lang I, Byng R, et al. What approaches to social prescribing work, for whom, and in what circumstances? A realist review. *Health Soc Care Community* [Internet]. 2020 Mar 9;28(2):309–24.
5. Wong G, Greenhalgh T, Westthorp G, Buckingham J, Pawson R. RAMESES publication standards: realist syntheses. *BMC Med* [Internet]. 2013 [cited 2018 Jun 4];11(21).
6. Higgins J, Thomas J. *Cochrane Handbook for Systematic Reviews of Interventions*. Version 6.1 [Internet]. 2020
7. Higgins J, Lasserson T, Chandler J, Tovey D, Thomas J, Flemyng E, et al. Methodological Expectations of Campbell Collaboration Intervention Reviews (MECCIR).
8. Moore GF, Audrey S, Barker M, Bond L, Bonell C, Hardeman W, et al. Process evaluation of complex interventions: Medical Research Council guidance. *BMJ* [Internet]. 2015 Mar 19 [cited 2018 Sep 12];350(mar19 6):h1258–h1258.
9. Malterud K, Siersma VD, Guassora AD. Sample Size in Qualitative Interview Studies. *Qual Health Res* [Internet]. 2016 Nov 10 [cited 2018 Oct 28];26(13):1753–60.

VERSION 2 – REVIEW

REVIEWER	Dr Josephine Wildman Newcastle University, United Kingdom
REVIEW RETURNED	25-Nov-2020
GENERAL COMMENTS	The authors have taken the time to thoughtfully and thoroughly address my comments. This study will make a valuable contribution to the social prescribing evidence base and I look forward to reading about the findings.